# Zfhx3 Transcription Factor Represses the Expression of *SCN5A* Gene and Decreases Sodium Current Density (I_Na_)

**DOI:** 10.3390/ijms222313031

**Published:** 2021-12-02

**Authors:** Marcos Rubio-Alarcón, Anabel Cámara-Checa, María Dago, Teresa Crespo-García, Paloma Nieto-Marín, María Marín, José Luis Merino, Jorge Toquero, Rafael Salguero-Bodes, Juan Tamargo, Jorge Cebrián, Eva Delpón, Ricardo Caballero

**Affiliations:** 1Department of Pharmacology and Toxicology, School of Medicine, Universidad Complutense de Madrid, Instituto de Investigación Gregorio Marañón, CIBERCV, 28040 Madrid, Spain; marcru02@ucm.es (M.R.-A.); ancamara@ucm.es (A.C.-C.);; tcresp01@ucm.es (T.C.-G.); paloma.nieto.marin@ucm.es (P.N.-M.); mmarin08@ucm.es (M.M.); jtamargo@med.ucm.es (J.T.); edelpon@med.ucm.es (E.D.); rcaballero@med.ucm.es (R.C.); 2Department of Cardiology, Hospital Universitario La Paz, Instituto de Investigación Sanitaria la Paz, CIBERCV, 28046 Madrid, Spain; jlmerino@arritmias.net; 3Department of Cardiology, Hospital Universitario Puerta de Hierro, Instituto de Investigación Sanitaria Puerta de Hierro-Segovia de Arana, CIBERCV, Majadahonda, 28222 Madrid, Spain; jorgetoquero@hotmail.com; 4Department of Cardiology, Hospital Universitario 12 de Octubre, Instituto de Investigación Hospital 12 de Octubre, CIBERCV, 28041 Madrid, Spain; rafael.salguero@salud.madrid.org

**Keywords:** Zfhx3, *SCN5A*, Nav1.5, Tbx5, Pitx2c, cardiac, sodium current, patch-clamp

## Abstract

The *ZFHX3* and *SCN5A* genes encode the zinc finger homeobox 3 (Zfhx3) transcription factor (TF) and the human cardiac Na^+^ channel (Nav1.5), respectively. The effects of Zfhx3 on the expression of the Nav1.5 channel, and in cardiac excitability, are currently unknown. Additionally, we identified three Zfhx3 variants in probands diagnosed with familial atrial fibrillation (p.M1260T) and Brugada Syndrome (p.V949I and p.Q2564R). Here, we analyzed the effects of native (WT) and mutated Zfhx3 on Na^+^ current (I_Na_) recorded in HL-1 cardiomyocytes. *ZFHX3* mRNA can be detected in human atrial and ventricular samples. In HL-1 cardiomyocytes, transfection of Zfhx3 strongly reduced peak I_Na_ density, while the silencing of endogenous expression augmented it (from −65.9 ± 8.9 to −104.6 ± 10.8 pA/pF; *n* ≥ 8, *p* < 0.05). Zfhx3 significantly reduced the transcriptional activity of human *SCN5A*, *PITX2*, *TBX5*, and *NKX25* minimal promoters. Consequently, the mRNA and/or protein expression levels of Nav1.5 and Tbx5 were diminished (*n* ≥ 6, *p* < 0.05). Zfhx3 also increased the expression of Nedd4-2 ubiquitin-protein ligase, enhancing Nav1.5 proteasomal degradation. p.V949I, p.M1260T, and p.Q2564R Zfhx3 produced similar effects on I_Na_ density and time- and voltage-dependent properties in WT. WT Zfhx3 inhibits I_Na_ as a result of a direct repressor effect on the *SCN5A* promoter, the modulation of Tbx5 increasing on the I_Na_, and the increased expression of Nedd4-2. We propose that this TF participates in the control of cardiac excitability in human adult cardiac tissue.

## 1. Introduction

*ZFHX3* gene encodes Zfhx3 or the AT motif binding factor (ATBF1), a transcription factor (TF) with multiple homeodomains and zinc finger motifs. Zfhx3 is widely expressed in many tissues [1] and participates in the regulation of myogenic [2] and neuronal differentiation. Zfhx3 was reported to inhibit cell proliferation, negatively regulate c-Myb, and trans-activate the cell cycle and cyclin-dependent kinase inhibitor 1A, thus functioning as a tumor suppressor in several cancers [3]. Additionally, it was demonstrated that Zfhx3 participated in some TF networks in the heart. Indeed, Huang and coworkers showed that Zfhx3 positively and reciprocally regulated the expression of *PITX2*, which encodes the Pitx2c TF [4]. Moreover, both, Zfhx3 and Pitx2c, regulate the expressions of *NPPA*, *TBX5* and *NKX25* genes which encode the atrial natriuretic peptide, as well as Tbx5 and Nkx2.5 TFs, respectively [4].

Genome-wide association studies (GWAS) significantly associated atrial fibrillation (AF) with two variants (rs7193343 and rs2106261) in the *ZFHX3* gene that appear outside coding regions [5,6]. The rs7193343 variant was also associated with ischemic and cardioembolic stroke [7]. The association of the rs2106261 variant with AF was replicated in an Asian cohort. Moreover, rs2106261 was also associated with coronary disease in an African American cohort [8,9]. Previous studies did not identify a clear association between this SNP and measures of atrial structure [10]. Moreover, an update meta-analysis demonstrated that rs7193343 and rs2106261 were not associated with AF recurrence [11]. More recently, variants in *ZFHX3* were also associated with sick sinus syndrome [12]. 

The *SCN5A* gene encodes the human cardiac Na^+^ channel (Nav1.5), which generates the fast Na^+^ current (I_Na_). The I_Na_ is responsible for the Na^+^ influx that depolarizes the membrane potential during the atrial and ventricular action potential upstroke. Thus, it plays a critical role in excitability and intracardiac conduction velocity. Recently, it was functionally demonstrated that Tbx5, a TF belonging to the T-box family, promotes the expression of the *SCN5A* gene, and thus increased I_Na_ in human cardiomyocytes derived from induced pluripotent stem cells (hiPSC-CM) [13]. Furthermore, the Brugada Syndrome (BrS)-associated variant p.F206L Tbx5, lacks this pro-transcriptional effect, thus markedly reducing I_Na_ [13]. On the other hand, Nav1.5 expression decreases in atrial chamber-specific Pitx2 conditional mutants, suggesting that Pitx2c positively modulates *Scn5a* expression [14]. However, in another report it was demonstrated that *Scn5a* mRNA expression increases in the left atria of mouse after Pitx2 heterozygous deletion [15].

Here, we decided to analyze the effects of native (WT) Zfhx3 on the I_Na_ magnitude and the expression of the human minimal *SCN5A* promoter. Moreover, we also tested the effects of three different variants that we identified in unrelated probands diagnosed with familial AF (p.M1260T) and BrS (p.V949I and p.Q2564R). Our results demonstrated that WT Zfhx3 inhibits I_Na_ as a consequence of a direct repressor effect on the *SCN5A* promoter, by the modulation of Tbx5-increasing effects on the I_Na_, and by increasing the expression of Nedd4.2. Additionally, we described that all three of these variants also inhibit I_Na_ similarly to WT Zfhx3, even when p.Q2564R Zfhx3 lacks the repressor effect at the level of the minimal *SCN5A* promoter.

## 2. Results

### 2.1. Zfhx3 Is Expressed in the Human Myocardium

First, we questioned whether Zfhx3 was expressed in the human adult myocardium. To answer this question, we accessed the Genotype-Tissue Expression (GTEx) project which collects and analyzes multiple human post mortem tissues [16]. GTEx RNA-seq data of *ZFHX3* from human atrial (*n* = 297) and ventricular (*n* = 303) samples averaged 2.2 ± 0.05 and 1.4 ± 0.04 transcripts per million (TPM), respectively. These data suggest that Zfhx3 is indeed expressed, even though the mRNA expression level, both in the atria and ventricles, is significantly lower (*p* < 0.01) than that of *TBX5* (59 ± 1.5 and 12.6 ± 0.5 TPM in atria and ventricles, respectively) and *NKX25* (115 ± 3.3 and 106 ± 3.5 TPM). For comparison, mRNA expression levels of *PITX2* are significantly (*p* < 0.01) lower than those of *ZFHX3* (0.4 ± 0.09 and 0.02 ± 0.002 TPM in atria and ventricles, respectively). Certainly, Pitx2c is almost not expressed in human ventricles, while its low expression in the atria increases in patients with chronic AF [17]. 

### 2.2. Zfhx3 Markedly Reduces I_Na_

Figure 1A shows I_Na_ traces recorded in HL-1 cells transfected/not transfected with the plasmid encoding the native form of the human Zfhx3 using the protocol shown in the upper part. In Figure 1B, the I_Na_ density is represented as a function of the membrane potential of the test pulse. In cells transfected with Zfhx3, the I_Na_ density consistently and significantly decreased compared with non-transfected cells at several membrane potentials. In fact, maximum I_Na_ decreased from −70.6 ± 6.8 to −27.7 ± 2.8 pA/pF (*p* < 0.05, *n* ≥ 26). The overexpression of Zfhx3 in HL-1 cells was confirmed by WB (Figure 1E).

In another group of experiments, the expression of Zfhx3 in HL-1 cells decreased when using siRNAs (Figure 1F,G). The specificity of the silencing effects was confirmed by using a scrambled siRNA in “control” cells. Figure 1C,D shows that I_Na_ density was significantly increased in Zfhx3-silenced cells (*p* < 0.05, *n* ≥ 9). 

We also analyzed the possible effects of Zfhx3 on the voltage dependence of I_Na_ activation (Figure 2A,B) and inactivation (Figure 2C,D). Neither transfection nor silencing of Zfhx3 modified either the voltage dependence of activation or inactivation of the I_Na_. Consequently, the midpoint and slope values of the activation and inactivation curves were not modified under any experimental condition (Table 1) (*p* > 0.05, *n* ≥ 6).

Time dependence of I_Na_ activation was quantified by fitting a monoexponential function to the activation phase of the maximum I_Na_ trace of each experiment [13]. The activation time constant (τ_act_) averaged 0.19 ± 0.01 (*n* = 43) and 0.22 ± 0.03 ms (*n* = 19) in cells transfected/not transfected with Zfhx3, respectively, indicating that the TF did not modify the activation kinetics (*p* > 0.05) (Table 1). The inactivation kinetics of maximum I_Na_ current traces were described by a biexponential function. Figure 3A shows the fast (τ_f_) and slow (τ_s_) time constants of inactivation of peak I_Na_ recorded in HL-1 cells transfected/not transfected with Zfhx3. As can be observed, Zfhx3 expression did not modify the inactivation kinetics of the I_Na_ (Table 1). Consistently with these findings, Zfhx3 silencing did not modify it either (Figure 3B and Table 1). The persistent component of I_Na_ or late I_Na_ (I_Na,L_) was quantified as the percentage of the peak I_Na_ [13] and plotted in Figure 3C,D. These figures demonstrate that I_Na,L_ magnitude was not different in HL-1 cells transfected/not transfected with Zfhx3 (Figure 3C) (*p* > 0.05, *n* ≥ 14) or in cells in which Zfhx3 expression was silenced/not silenced (Figure 3D) (*p* > 0.05, *n* ≥ 7). Finally, we also analyzed the reactivation kinetics of the I_Na_ using a double-pulse protocol (upper panel in Figure 3E). The reactivation process was described by the fit of a monoexponential function to the data and, as is shown in Figure 3E and Table 1, Zfhx3 did not significantly modify this (*p* > 0.05, *n* ≥ 11). The reactivation kinetics were also identical in cells in which the expression of Zfhx3 was silenced/not silenced by means of specific siRNAs (*p* > 0.05, *n* ≥ 4) (Figure 3F and Table 1).

In conclusion, Zfhx3 markedly decreased I_Na_ without modifying its voltage- and time-dependent characteristics. These data suggested that Zfhx3 inhibited I_Na_ just by decreasing the expression of the Nav1.5 proteins. Thus, Western blot (WB) experiments were conducted to test this hypothesis and demonstrated that HL-1 cells transfected with Zfhx3 significantly decreased Nav1.5 protein levels (*p* < 0.05, *n* ≥ 7) (Figure 4A,B). Moreover, a concomitant decrease in the *Scn5a* mRNA expression was detected by Reverse Transcription Quantitative PCR (RT-qPCR) experiments (*p* < 0.05, *n* ≥ 4) (Figure 4C). Finally, we tested whether Zfhx3 directly reduced the expression of the human minimal *SCN5A* promoter using luciferase assays. Indeed, Zfhx3 repressed the expression of the minimal *SCN5A* promoter, as well as that of the human minimal *SCN1B* promoter, which encoded the Navβ1 ancillary subunit of the cardiac Na^+^ channel (*p* < 0.05, *n* ≥ 3) (Figure 4D).

### 2.3. Zfhx3, Tbx5 and Pitx2c Interplay

Previous results suggested that there is a positive reciprocal modulation between *ZFHX3* and *PITX2C* [4]. Additionally, both Zfhx3 and Pitx2c regulate the expression of other cardio-specific TF such as Tbx5 and Nkx2.5 [4]. Thus, we decided to test the effect of Zfhx3 on the expression of these TFs, which also regulated the magnitude of cardiac I_Na_ [13]. Figure 5A,B shows that transfection with Zfhx3 significantly decreased the protein expression of Tbx5 as demonstrated by WB experiments. Moreover, Zfhx3 significantly decreased the *TBX5* mRNA expression (Figure 5C) (*p* < 0.05, *n* = 6), an effect that we attributed to the remarkable inhibition produced by Zfhx3 on the expression of the human minimal *TBX5* promoter as detected in luciferase assays (Figure 5D) (*p* < 0.01, *n* = 6). Regarding Pitx2c, unfortunately the HL-1 expression of both the protein and the mRNA was negligible; therefore, we were not able to accurately detect them with the antibodies and primers used in WB and RT-qPCR experiments (see Methods in the Appendix A). Conversely, in luciferase assays we could demonstrate that Zfhx3 significantly repressed the expression of the human minimal *PITX2* promoter (Figure 5D) (*p* < 0.01, *n* = 4). Finally, the results of the luciferase assay demonstrated that Zfhx3 also markedly and significantly decreased the expression of the human minimal *NKX25* promoter (Figure 5D) (*p* < 0.01, *n* = 3).

### 2.4. Zfhx3 and Nedd4.2 Expression

It was extensively demonstrated that the ubiquitin protein ligase Nedd4-2 ubiquitinates Nav1.5 by binding to the PY motif located at the C-terminus of the channel, and promotes its degradation by the proteasome [18]. Since the expression of Zfhx3 markedly reduced I_Na_, we questioned whether it promoted in some way, the degradation of Nav1.5 channels. The WB data shown in Figure 6A,B demonstrated that the protein expression of Nedd4.2 significantly increased in HL-1 cells transfected with Zfhx3 (*p* < 0.05, *n* = 6). In accordance, mRNA levels of *Nedd4l* also significantly increased (*p* < 0.05, *n* = 7) (Figure 6C).

### 2.5. p.V949I, p.M1260T, and p.Q2564R Zfhx3 Variants

The first proband (BrS-1; III:1) is a 38-year-old male diagnosed with BrS (Figure 7A,B) who carries an implantable cardioverter defibrillator. His mother (II:3), uncle (II:1), aunt (II:5, mother’s sister), and sister (III:2) died suddenly at the ages of 59, 50, 51, and 32, respectively (Figure 7A). Another proband’s uncle (II:2) suffered a sudden cardiac arrest at age 72, but he was resuscitated. Gene panel sequencing identified that his mother carried a rare variant in *KCNH*2 encoding p.Leu839Pro human ether-a-go-go (hERG) channels that generate the rapid component of the delayed rectifier current (I_Kr_). The p.L839P hERG variant was never annotated in the Genome Aggregation Database (gnomAD, https://gnomad.broadinstitute.org/, accession on 29 November 2021) nor related with BrS. The gene panel sequencing demonstrated that the proband did not carry this variant (Table 2). However, it revealed that the patient carried a rare variant in *CACNA1*B gene encoding p.Arg377Gln Cav2.2 channels that was predicted as deleterious by two out six of the software tools for prediction of the impact of amino acids substitution (Table 2). The p.R377Q Cav2.2 variant, in turn, was not present in the patient’s mother (Figure 7A). No genetic data were available from his aunt and sister. The proband has two cousins (III:3 and III:4; daughters of his deceased aunt), who also carry the p.R377Q Cav2.2 variant and two nephews (IV:1 and IV:2; daughters of his deceased sister) who do not carry *KCNH2* or *CACNA1B* variants (Figure 7A). His cousins and nephews exhibit normal ECGs and do not have history of syncope. Furthermore, his cousins and another aunt (II:4) underwent a flecainide test that was negative. 

Next-generation sequencing of the proband identified the p.Val949Ile Zfhx3 variant (rs113497421) that was predicted as deleterious (Table 2). The proband does not carry any other variant predicted as pathogenic (according to the guidelines for the interpretation of variants, see Appendix A) [19] in genes linked with BrS. Considering the effects of Zfhx3 on I_Na_, we hypothesized that the p.V949I variant could act as a genetic modifier and we decided to analyze its presence in the relatives available for a genetic test by Sanger sequencing. None of them carried the p.V949I Zfhx3 variant (Figure 7A).

We also identified two 59- and 61-year-old siblings with AF (AF-1 and AF-2) (Figure 7C). AF-1 exhibited paroxysmal AF (Figure 7D) with frequent episodes that were suppressed when the patient underwent catheter ablation. One year later she was ablated to eliminate a right ventricular outflow tract tachycardia. At the present time, she suffers sporadic palpitations of a short duration (<1 min). AF-2 had persistent AF refractory to multiple electrical cardioversions that progressed to permanent AF treated with amiodarone. Additionally, he had hypertrophic cardiomyopathy as demonstrated by echocardiography. Their deceased mother (I:2) also had AF diagnosed when she was 50 years old, and suffered two stroke episodes. The deceased father (I:1) and sister II:1 of the probands also exhibited hypertrophic cardiomyopathy, while II:4 died of brain cancer when she was 55 years old. The two index cases have children and nephews who have not presented with any electrical or structural alterations to date (third generation; Figure 7C). Next-generation sequencing of the probands identified that they shared the p.Met1260Thr Zfhx3 variant, predicted as deleterious by two out of six prediction tools (Table 2). The presence of this variant was analyzed by Sanger sequencing in other members of the family, and was identified in one of the daughters (III:6) of AF-2 (Figure 7C). 

The third proband (II:3, BrS-2) was a 59-year-old woman who was diagnosed with BrS by an ECG conducted for pre-surgical purposes (Figure 7E,F). Thereafter, she underwent a flecainide test that was positive. She was and remains asymptomatic; however, her cousin (II:1, Figure 7E) suddenly died in his sleep when he was 49 years old. Her son (III.1) and daughter (III.2) exhibit normal ECG and had negative flecainide tests. Next-generation sequencing of the proband identified the p.Gln2564RArg Zfhx3 and the p. Asp3126Gly Ankyrin B variants. The latter was predicted as deleterious (Table 2). Furthermore, *ANK2* loss-of-function variants were extensively related with inherited cardiac arrhythmias [20], even with BrS [21], although this relationship is currently under debate [22]. 

### 2.6. Effects of the p.V949I, p.M1260T, or p.Q2564R Zfhx3 on the I_Na_

To functionally test the effects of these variants on I_Na_ we used HL-1 cells transfected/not transfected with p.V949I, p.M1260T, or p.Q2564R Zfhx3. As can be observed in Appendix A, all these residues are very conserved in different species. The *ZFHX3* mRNA expression is not different upon the transfection of cells with the cDNA encoding WT or mutated Zfhx3, as demonstrated by qPCR experiments (Cycle to threshold = 25.9 ± 0.3, 25.7 ± 0.3, 26.1 ± 0.2, and 25.6 ± 0.3 for WT, p.V949I, p.M1260T, and p.Q2564R Zfhx3, *n* = 7, *p* > 0.05) (Appendix A). The I_Na_ density recorded in cells transfected with these mutants was not statistically different than that recorded in cells transfected with the WT form (*p* > 0.05, *n* ≥ 17) (Figure 8A). Therefore, as in the presence of WT Zfhx3, maximum I_Na_ density was markedly and significantly decreased in cells transfected with Zfhx3 p.V949I, p.M1260T, or p.Q2564R Zfhx3 compared with non-transfected cells (*p* > 0.01, *n* ≥ 17) (Figure 8B). As was the case with the WT form, none of the variants significantly modified any time- or voltage-dependent characteristics of the I_Na_ (Table 1). 

Considering these results, we assumed that all three variants retained their repressor activity in the human minimal *SCN5A* and *SCN1B* promoters. Figure 8C shows that p.V949I and p.M1260T Zfhx3 significantly repressed *SCN5A* promoter expression (*p* < 0.01, *n* = 3), whereas p.Q2564R produced no such effect. Conversely, Figure 8D shows that all three variants significantly repressed the expression of the *SCN1B* promoter (*p* < 0.01, *n* = 3). 

Since p.Q2564R Zfhx3 did not repress the expression of the *SCN5A* promoter, the question arose of how this variant also reduced the I_Na_ density. To answer it we conducted luciferase assays with all three Zfhx3 variants to test their effects on the expression of the human minimal *TBX5*, *PITX2* and *NKX25* promoters. Figure 9A shows that all three variants, including p.Q2564R, significantly repressed the expression of the minimal *TBX5* promoter (*p* < 0.01, *n* ≥ 3). p.V949I and p.M1260T also significantly repressed the expression of the minimal *PITX2* and *NKX25* promoters (Figure 9B,C) (*p* < 0.01, *n* ≥ 3). Conversely, p.Q2564R Zfhx3 failed to repress the expressions of both *PITX2* and *NKX25* promoters (Figure 9B,C) (*p* < 0.01, *n* ≥ 3). 

## 3. Discussion

Our results demonstrate that Zfhx3 inhibits I_Na_ as a result of a direct repressor effect on the *SCN5A* promoter, as well as the increased expression of Nedd4-2, and the modulation of Tbx5-increasing effects on the I_Na_. Thus, we provide a novel and complex mechanism by which this TF could modulate cardiac excitability. 

The overexpression of the cDNA encoding Zfhx3 in HL-1 cells markedly reduced I_Na_ density, while Zfhx3 silencing led to the opposite result. Neither expression nor silencing modified the time- and voltage-dependent properties of the current or the I_Na,L_, suggesting that the factor affected mostly the expressions of channel proteins, rather than gating. Our luciferase experiments demonstrated that Zfhx3 markedly decreases transcriptional activity of human *SCN5A* and *SCN1B* minimal promoters. Zfhx3 regulates transcription via direct interactions with predicted adenine and thymine-rich (AT) motifs [23] and these motifs are present in the promoter regions of both *SCN5A* and *SCN1B* genes (see Appendix A). The effects on the *SCN5A* promoter were correlated with a reduction in mRNA and the protein levels of Nav1.5 channels. With regard to Navβ1, its effects on I_Na_ properties depend on the expression system used [24]. Nevertheless, it is generally accepted that it increases the Nav1.5 cell surface expression [25]. Thus, it is possible that the decrease in Navβ1 levels produced by Zfhx3 contributes to I_Na_ inhibition. Zfhx3 controls development, tumorigenesis and other biological processes that require a fast and dynamic protein turnover [26]. Therefore, we hypothesized that Zfhx3 could affect Nav1.5 channel degradation, in addition to synthesis. Indeed, we demonstrated that Zfhx3 increased the Nedd4-2 expression, an effect that favoured the proteasomal degradation of the channels [18] and would contribute to the net Zfhx3-induced I_Na_ inhibition.

To the best of our knowledge, there are no previous data demonstrating a regulatory role of Zfhx3 on Nav1.5 channels. Conversely, data showing its effects on the expression and function of cardiac K^+^ channels are more abundant [4,27,28]. It was shown that Zfhx3 silencing in HL-1 cells increased the expressions of Kv1.4, Kv1.5 and Kir3.4 channels, which resulted in augmented ultra-rapid delayed rectifier (I_Kur_), transient outward (I_to_) and acetylcholine-sensitive potassium (I_KAch_) currents and the shortening of the action potential duration [27]. More recently, Lkhagva et al. described that Zfhx3 silencing in HL-1 cells led to a significant increase in the ATP-sensitive K current (I_KATP_) [28]. There are also data suggesting that Zfhx3 regulates intracellular calcium handling, since Zfhx3 knockdown in HL-1 cells increased the ryanodine receptor (*RYR2*), RyR2 p2808, and SERCA2a (*ATP2A2*) mRNA expression [27].

There is evidence showing that Zfhx3 establishes gene regulatory networks with other cardio-enriched TFs such as Tbx5 and Pitx2c [29], suggesting that these TFs could regulate the expression of genes relevant to cardiac electrical activity in a coordinated manner. Indeed, Huang and coworkers demonstrated that, in HCT116 cells, Pitx2c negatively regulates the expression of miR-1, which reduced the expression of *ZFHX3*, resulting in a positive regulation of *ZFHX3* by Pitx2c. Zfhx3, in turn, positively regulates expression of *PITX2*, resulting in a cyclic loop of cross-regulation between *ZFHX3* and *PITX2* [4]. Both, Zfhx3 and Pitx2c, regulate the expressions of *NPPA*, *TBX5* and *NKX25* genes which encoded the atrial natriuretic peptide, and Tbx5 and Nkx2.5 TF, respectively. Additionally, Tbx5 may also regulate the expression of Zfhx3 [30]. Furthermore, other TFs could also be involved. Indeed, a gene interaction network dominated by Nkx2-5, Tbx3, Zfhx3, and Synpo2l was recently identified in the human left atria [31].

Our luciferase assays, RT-qPCR and WB experiments confirmed the results of Tbx5, since the expression of Zfhx3 reduces the activity of the minimal promoter and the mRNA and protein levels of Tbx5. On the other hand, we recently demonstrated that Tbx5 increases Nav1.5 channel expression and I_Na_ density in hiPSC cardiomyocytes by means of a pro-transcriptional effect on the gene promoter [13]. Thus, we assume that the repression of the *TBX5* gene transcription produced by Zfhx3 contributes to the I_Na_ inhibition produced by Zfhx3.

Conversely, under our experimental conditions, the expression of Zfhx3 did not increase, but decreased the luminescence generated by the human *PITX2* minimal promoter. Unfortunately, in HL-1 cells, we were not able to detect measurable mRNA and protein levels of Pitx2c to confirm these results, most likely due to a limited expression of this factor. The reasons underlying the discrepancy between our results and those of Huang and coworkers [4] are unknown, although the use of different cellular models could play a role. The biology and expression levels of Zfhx3 and Pitx2c in HL-1 cells, which are of murine cardiac origin, and HCT116 cells, which are of human colon cancer origin, are likely very different. 

Contradictory results were obtained regarding the effects of Pitx2c and the expression of the *Scn5a* gene. Previous reports demonstrated that Nav1.5 expression decreased in atrial-chamber-specific Pitx2 conditional mutants [14], while it increased in mouse left atria after *Pitx2* heterozygous deletion [15]. Thus, we cannot predict the net effect on the I_Na_ magnitude produced by the repressor effects of Zfhx3 on the expression of the *PITX2* gene.

We also showed that Zfhx3 decreased the luciferase activity generated by the human *NKX25* promoter. Our results suggest that Zfhx3, Tbx5, Pitx2c, and Nkx2-5 TFs reciprocally interact with each other in a complex way to control the expression of Nav1.5 channels, and thus cardiac conduction velocity and excitability. Furthermore, changes in the expression levels of these TFs would affect the I_Na_ magnitude. In this sense, it was proposed that noncoding variations in regulatory sequences affect the expression of genes encoding transcriptional regulators and/or the function of regulatory elements of given target genes, altering gene expression and conferring disease susceptibility [29]. Should this occur, changes in Zfhx3 levels produced by the SNPs identified in GWAS studies disrupted the equilibrium within the transcriptional networks, leading to important consequences on cardiac excitability.

We also analyzed the consequences of three *ZFHX3* variants found in probands that did not carry any variant in the genes so far associated with inherited arrhythmogenic syndromes. Our results show that the three variants produced a similar reduction in the I_Na_ density as the WT form, suggesting that they do not cause familial AF or BrS through differential effects on I_Na_. It is possible that these variants contribute to the phenotype of the carriers through different mechanisms not explored here, such as affecting intracellular calcium-handling proteins [27], changing the expression of *ZFHX3* or other genes, or acting as genetic modulators of another still unidentified causative variant [32]. Moreover, as described in Table 2, all three patients also carry other nonsynonymous exonic missense variants predicted as potentially deleterious. It is possible, that the phenotype of the patients is only apparent when, in addition to the variant in *ZFHX3*, the other variants of which they are carriers are present. Unfortunately, our experiments in HL-1 cells cannot rule out this hypothesis since it would be necessary to generate cardiomyocytes derived from iPSC from each patient and to conduct the sequential correction of each variant with CRISPR-Cas9 technology. Tsai et al. identified four missense variants (p.E460Q, p.V777A, p.M1476I, and p.S3513G) in the coding region of the *ZFHX3* gene in patients with AF, although they did not functionally analyze the consequences of the variants [33]. To compare the predicted risk of these AF-associated variants with the variants identified here, we obtained the Combined Annotation Dependent Depletion (CADD) score (https://cadd.gs.washington.edu/, accession on 29 November 2021). CADD considers a wide range of functional categories, effect sizes and genetic architectures, and can be used to prioritize causal variations [34]. The score predicts that the risk of p.V949I and p.Q2564R is slightly higher than p.E460Q and much higher than the rest of the variants (Appendix A). Our luciferase assays showed that the p.Q2564R variant exhibited a different transcriptional profile from WT and the other variants analyzed here, since it did not reduce the luciferase activity generated by *SCN5A* and *PITX2* promoters. The reason for these differences could be related to the position of the variant. Both p.V949I and p.M1260T are within the eighth and twelfth zinc finger domains, while p.Q2564R is the closest to a DNA-binding domain, being in a reasonable proximity to the third homeodomain of Zfhx3 (2641-2700, NP_008816.3). Although the p.Q2564R variant led to a similar reduction in I_Na_ density than WT and the rest of the variants, our data suggest that it reduced I_Na_ density by suppressing the Tbx5-increasing effects, without producing a direct effect on *SCN5A* gene transcription. 

### Limitations of the Study

We did not analyze the effects of Zfhx3 expression on I_Na_ recorded in cultured native cardiomyocytes. To this end, we would need to include the cDNA of Zfhx3 within some viral vector. Unfortunately, the size of *ZFHX3* (17,669 bp) prevents its inclusion in the vast majority of available viral vectors. We used HL-1 cells that were of a murine cardiac origin and were widely used for these purposes as a good model of cardiomyocytes in culture [35]. However, since HL-1 cells are not differentiated enough and do not display lateral membranes and intercalated disks, as is the case for adult cardiomyocytes, we were not able to analyze the possible effects of Zfhx3 on Nav1.5 localization. We surmise that Zfhx3 impacts the expression of the channels within the whole cardiomyocyte. However, we cannot rule out the differential effects on Nav1.5 targeted to specific sub-domains, since Zfhx3 may affect the expression/function of some of the multiple proteins that define distinct pools of Nav1.5 channels in cardiomyocytes [36,37]. Furthermore, the I_Na_ inhibition exceeds the decrease in the total expression of Nav1.5 protein produced by Zfhx3. This would suggest that Zfhx3 ultimately decreases the presence of Nav1.5 channels in the cell membrane (not explored here) by additional translational or post-translational mechanisms that are currently unknown. We did not demonstrate the binding of Zfhx3 to its target promoters by EMSA. Again, the huge size of the Zfhx3 protein (404 kDa) hampers its synthesis via conventional systems based on reticulocytes [13]. 

## 4. Material and Methods

### 4.1. Study Approval

The Investigation Committees of the University Hospitals La Paz, 12 de Octubre and Puerta de Hierro (ITACA study) approved the clinical evaluation of probands and all family members. Studies conformed to the principles outlined in the Declaration of Helsinki. Each patient gave written informed consent. 

### 4.2. DNA Sequencing

DNA was extracted from whole blood, and whole-exome sequencing was performed at NIMgenetics (Madrid, Spain) [13,32]. Appendix A describes the genes selected for the bioinformatics analysis performed after whole exome sequencing. The presence of the variants was also confirmed in the probands and some family members by Sanger sequencing.

### 4.3. Access to Public Human Cardiac RNA-Seq Data

GTEx RNA-seq data (RSEMv1.2.22 (v7) version) from human atria and ventricles and the resulting TPM values for *ZFHX3*, *TBX5*, *PITX2*, and *NKX25* were accessed through the Human Protein Atlas database (https://www.proteinatlas.org/, accession on 29 November 2021) [13].

### 4.4. Cell Culture and Transfection

HL-1 cells were cultured and transiently transfected with the cDNA-encoding human WT or mutant Zfhx3 by using Lipofectamine 2000 (Invitrogen, Carlsbad, CA, USA) as previously described [13,17,32].

### 4.5. Patch-Clamp Recordings

I_Na_ was recorded at room temperature using the whole-cell, patch-clamp technique [13,17,32,38]. Micropipette resistance was kept below 1.5 MΩ when filled with the internal solution and immersed in the external solution (see composition in Appendix A). To minimize the influence of the expression variability, each construct was tested in a large number of cells obtained from at least 3 different HL-1 batches. In all cases, the expression of WT or mutated Zfhx3 was identified by the green fluorescent signal under fluorescent microscopy.

### 4.6. Analysis of the mRNA Expression (RT-qPCR)

mRNA expression of *ZFHX3*, *SCN5A*, *SCN1B*, *NEDD4L*, *TBX5*, *PITX2*, and *NKX25* was measured by RT-qPCR using TaqMan Gene Expression Assays [17,32].

### 4.7. Western Blot Analysis and Zfhx3 Silencing

Nav1.5, Nedd4-2, and Tbx5 proteins were detected by WB in HL-1 cells transfected/not transfected with WT or mutated Zfhx3 following procedures previously described. For Zfhx3 silencing, HL-1 cells were transfected with ON-TARGETplus mouse Zfhx3 siRNA SMARTpool or with siRNA Universal Negative Control (scrambled) by using Lipofectamine 2000 (Invitrogen), according to the manufacturer’s instructions [13,17,32,36,38].

### 4.8. Luciferase Assays

HL-1 cells were transfected with pLightSwitch_Prom [Active Motif, Carlsbad) vectors carrying the minimal promoters of human *SCN5A*, *SCN1B*, *TBX5*, *PITX2*, and *NKX25* genes and luminescence was measured as described [13,17,32].

### 4.9. Statistical Analyses 

Throughout the paper results were expressed as mean ± SEM. Statistical analyses were performed using GraphPad Prism 8 (GraphPad Software, San Diego, CA, USA). To compare data from ≥3 experimental groups, one-way ANOVA followed by Tukey’s test was used, while unpaired two-sided *t*-test was chosen when comparing data from two experimental groups. In small size samples (n < 15), statistical significance was confirmed by using non-parametric tests (two-sided Wilcoxon’s test). To take into account repeated sample assessments, data were analyzed with multilevel mixed-effects models. Normality assumption was verified using the Shapiro–Wilk test. Variance was comparable between groups throughout the manuscript. We chose the appropriate tests according to the data distributions. A value of *p* < 0.05 was considered significant. Additional methodological details are included in Appendix A.

## 5. Conclusions

Zfhx3 inhibits I_Na_ by a complex mechanism involving the reduction in Nav1.5 channel expression, the modulation of the Tbx5-increasing effects, and the increase in Nav1.5 channel degradation by the proteasome. We propose that this TF could participate in the control of cardiac excitability in human adult cardiac tissue.

## Figures and Tables

**Figure 1 ijms-22-13031-f001:**
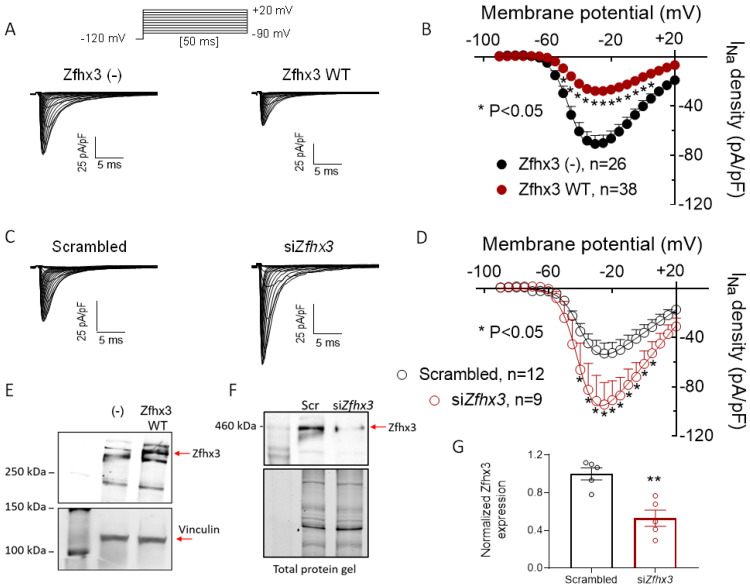
(**A**,**C**). I_Na_ traces recorded by applying the protocol shown at the top in HL-1 cells transfected/not transfected (–) with Zfhx3 WT, (**A**) or with scrambled or siRNA against Zfhx3 (si*Zfhx3*) (**C**). (**B**,**D**). Mean current-density voltage curves for I_Na_ recorded in HL-1 cells transfected/not transfected with Zfhx3 WT (**B**) or with scrambled or si*Zfhx3* (**D**). (**E**) Representative WB images showing the expression of Zfhx3 (top) and vinculin (bottom; loading control) in cells transfected/not transfected with Zfhx3 WT. (**F**,**G**). Representative WB (top) and total protein gel (bottom) images (**F**) and the corresponding densitometric analysis (**G**) showing the expression of Zfhx3 (red arrow) in cells transfected with Scrambled (Scr) or si*Zfhx3*. In (**B**,**D**,**G**) each point/bar is the mean ± SEM of “*n*” experiments. ** *p* < 0.01 vs. Scrambled. In (**B**,**D**), ANOVA followed by Tukey’s test and multilevel mixed-effects model; in (**F**) un-paired two tailed Student’s *t*-test and multilevel mixed effects model.

**Figure 2 ijms-22-13031-f002:**
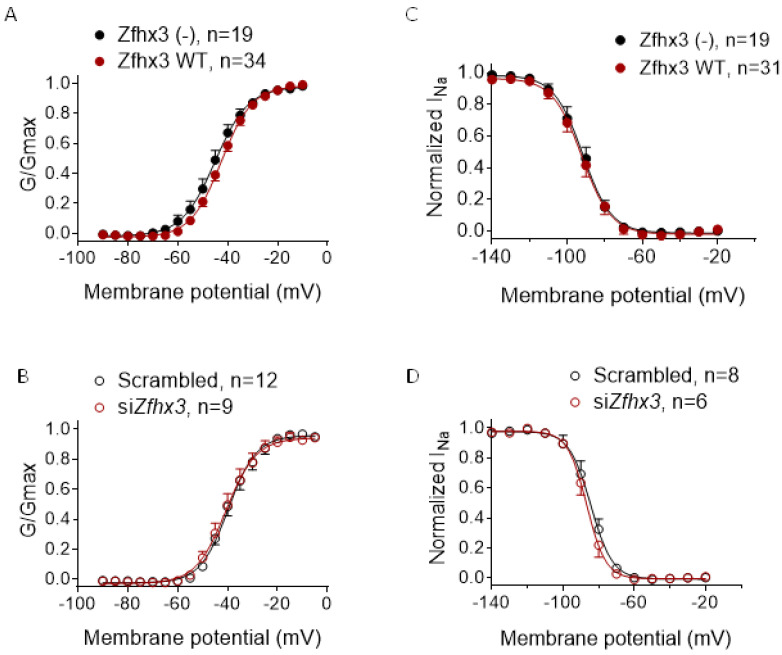
(**A**,**B**). Voltage-dependence of I_Na_ activation (G/Gmax) in HL-1 cells transfected/not transfected (–) with Zfhx3 WT (**A**) or with scrambled or siRNA against Zfhx3 (si*Zfhx3*) (**B**). (**C**,**D**). Voltage dependence of I_Na_ inactivation in HL-1 cells transfected/not transfected with Zfhx3 WT (**C**) or with scrambled or si*Zfhx3* (**D**). In (**A**–**D**) continuous lines represent the fit of a Boltzmann equation. Each point is the mean ± SEM of “*n*” experiments.

**Figure 3 ijms-22-13031-f003:**
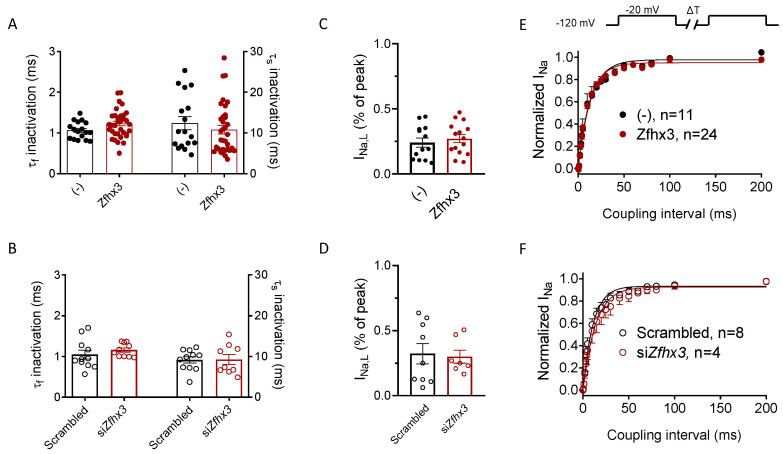
(**A**,**B**). Fast and slow time constants of inactivation obtained by fitting a biexponential function to the maximum I_Na_ traces in HL-1 cells transfected/not transfected with Zfhx3 WT (**A**), or with scrambled or siRNA against Zfhx3 (si*Zfhx3*) (**B**). (**C**,**D**). Mean I_NaL_ recorded in HL-1 cells transfected/not transfected with Zfhx3 WT (**C**), or with scrambled or si*Zfhx3* (**D**). (**E**,**F**). Time course of the recovery of I_Na_ inactivation in HL-1 cells transfected/not transfected with Zfhx3 WT (**E**) or with scrambled or si*Zfhx3* (**F**). Each point/bar is the mean ± SEM of “*n*” experiments.

**Figure 4 ijms-22-13031-f004:**
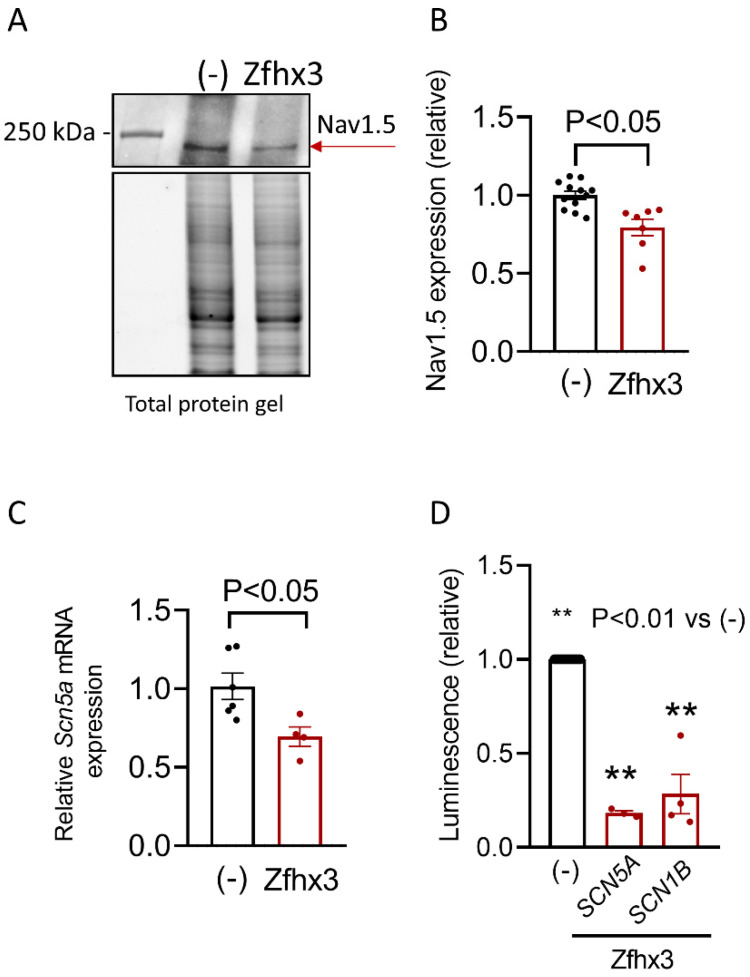
(**A**,**B**). Representative WB (top) and total protein gel (bottom) images (**A**) and densitometric analysis (**B**) of the expression of Nav1.5 (red arrow) in HL-1 cells transfected/not transfected with Zfhx3 (**B**). (**C**,**D**). *Scn5a* mRNA levels (**C**), and relative luminescence values generated by human *SCN5A* and *SCN1B* minimal promoters (**D**) measured in HL-1 cells transfected/not transfected with Zfhx3 WT. In (**B**–**D**), each bar is the mean ± SEM of “*n*” experiments; each dot represents 1 experiment in (**B**,**C**) and is the mean value of a technical triplicate in (**D**). (**B**,**C**) Un-paired, two-tailed Student’s *t*-test and multilevel mixed-effects model. (**D**) ANOVA followed by Tukey’s test and multilevel mixed-effects model.

**Figure 5 ijms-22-13031-f005:**
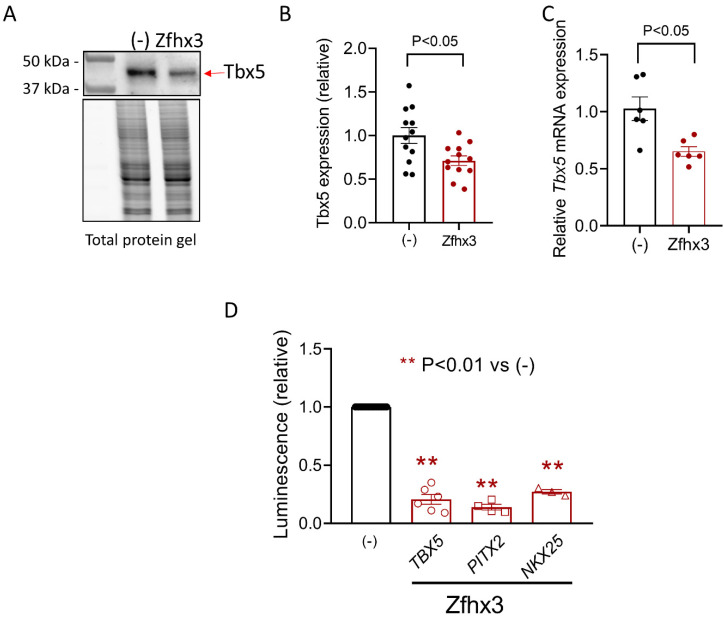
(**A**–**D**). Representative WB (top) and total protein gel (bottom) images (**A**); densitometric analysis of Tbx5 expression (red arrow) (**B**); *Tbx5* mRNA levels (**C**); and relative luminescence values generated by human *TBX5*, *PITX2*, and *NKX25* minimal promoters (**D**) measured in HL-1 cells transfected/not transfected with Zfhx3 WT. In (**B**–**D**), each bar is the mean ± SEM of “*n*” experiments; each dot represents 1 experiment in (**B**,**C**) and is the mean value of a technical triplicate in (**D**). (**B**,**C**) Un-paired two tailed Student’s *t*-test and multilevel mixed-effects model. (**D**) ANOVA followed by Tukey’s test and multilevel mixed-effects model.

**Figure 6 ijms-22-13031-f006:**
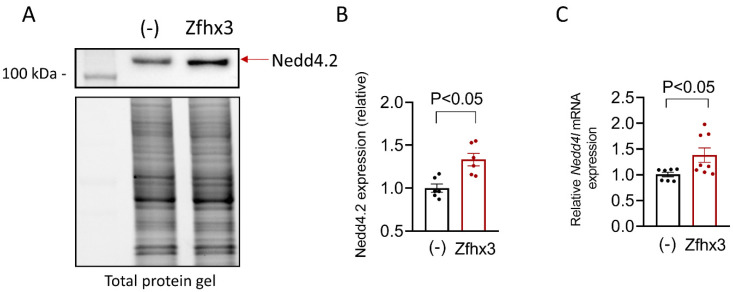
(**A**–**C**). Representative WB (top) and total protein gel (bottom) images (**A**), densitometric analysis of Nedd4-2 expression (red arrow) (**B**), and *Nedd4l* mRNA levels (**C**) measured in HL-1 cells transfected/not transfected with Zfhx3 WT. In (**B**,**C**), each bar is the mean ± SEM of “*n*” experiments and each dot represents 1 experiment. Un-paired two tailed Student’s *t*-test and multilevel mixed-effects model.

**Figure 7 ijms-22-13031-f007:**
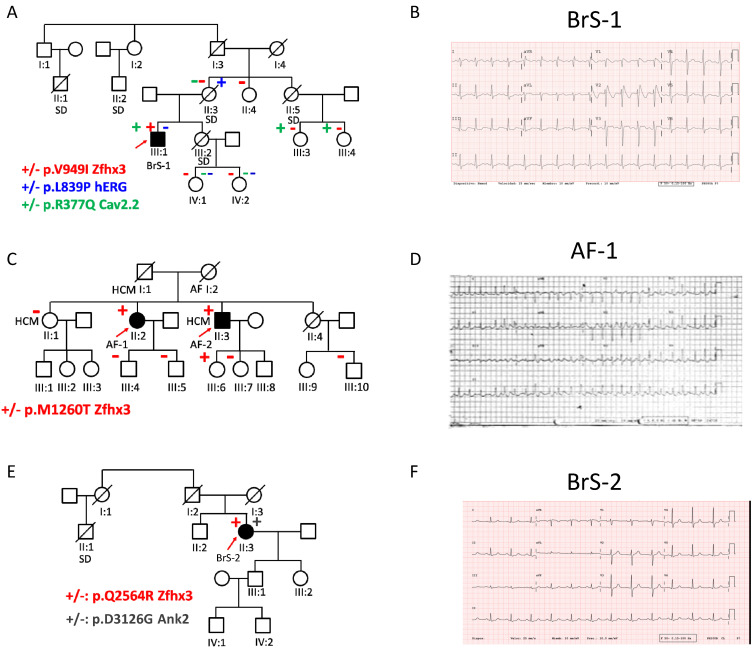
(**A**,**C**,**E**). Pedigrees of the three families of this study. The arrows indicate the probands; circles and squares represent females and males, respectively, and filled symbols represent the subjects diagnosed with BrS (**A**,**E**) or familial AF; (**C**) + or − indicate the presence or absence of the variants. AF: atrial fibrillation; BrS: Brugada Syndrome; HCM: hypertrophic cardiomyopathy; SD: sudden death. (**B**,**D**,**F**). Representative ECGs of BrS-1 (**B**), AF-1 (**D**) and BrS-2 (**F**) probands.

**Figure 8 ijms-22-13031-f008:**
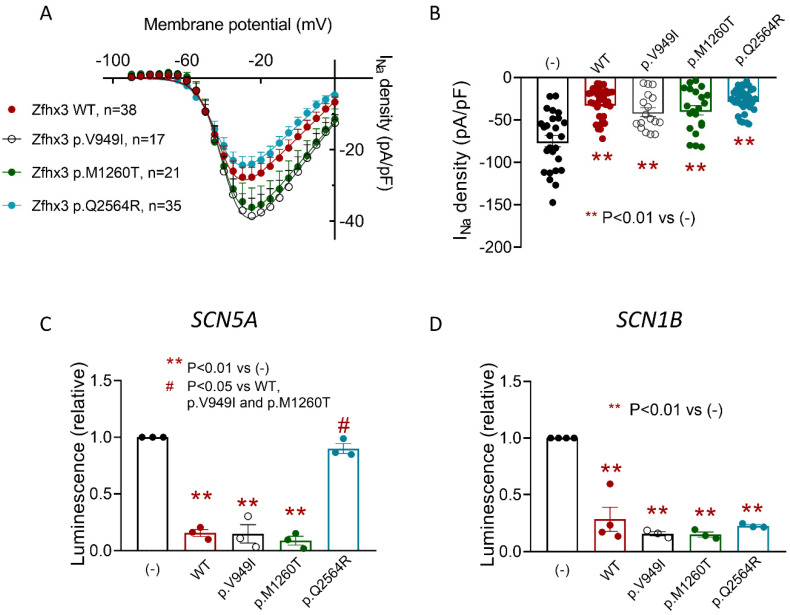
(**A**,**B**). Current-density voltage curves (**A**) and mean peak density (**B**) for I_Na_ recorded in HL-1 cells transfected with the indicated constructs. (**C**,**D**). Mean luminescence values measured in HL-1 cells transfected with human *SCN5A* (**C**) or *SCN1B* (**D**) minimal promoters together with the indicated Zfhx3 constructs. Each point/bar is the mean ± SEM of “*n*” experiments; in (**B**), each dot represents 1 experiment and, in (**C**,**D**), is the mean value of a technical triplicate. ANOVA followed by Tukey’s test and multilevel mixed-effects model.

**Figure 9 ijms-22-13031-f009:**
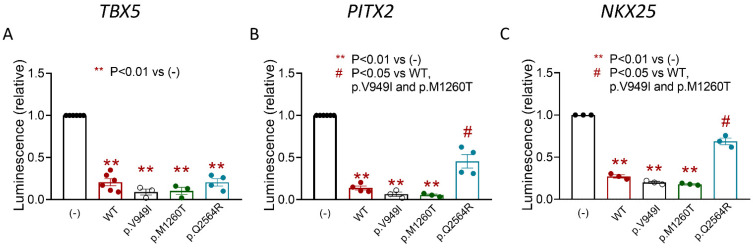
(**A**-**C**). Mean luminescence values measured in HL-1 cells transfected with human minimal *TBX5* (**A**), *PITX2* (**B**) or *NKX25* (**C**) promoters together with the indicated Zfhx3 constructs. Each bar is the mean ± SEM of “*n*” experiments and each dot represents the mean value of a technical triplicate. ANOVA followed by Tukey’s test and multilevel mixed-effects model.

**Table 1 ijms-22-13031-t001:** Effects on the time- and voltage-dependent properties of I_Na_.

Zfhx3	τ_act_ (ms)	V_hact_ (mV)	*k* _act_	τ_finact_ (ms) A_finact_ (%)	τ_sinact_ (ms) A_sinact_ (%)	V_hinact_ (mV)	*k* _inact_	τ_react_ (ms)
(–)	0.22 ± 0.03	−45.4 ± 1.9	5.1 ± 0.3	1.1 ± 0.05 85.2 ± 1.8	12.4 ± 1.6 14.8 ± 1.8	−91.8 ± 2.2	5.0 ± 0.1	14.9 ± 4.3
WT	0.19 ± 0.01	−42.6 ± 0.9	5.5 ± 0.2	1.2 ± 0.06 83.5 ± 1.8	10.8 ± 1.1 16.5 ± 1.8	−91.7 ± 1.8	5.2 ± 0.2	12.9 ± 1.4
p.V949I	0.20 ± 0.02	−40.8 ± 2.2	5.1 ± 0.3	1.1 ± 0.07 83.3 ± 1.8	11.7 ± 1.5 16.7 ± 1.8	−90.6 ± 2.0	5.1 ± 0.2	15.0 ± 2.2
p.M1260T	0.24 ± 0.04	−43.6 ± 2.3	5.3 ± 0.3	1.1 ± 0.1 80.9 ± 1.9	12.1 ± 1.1 19.1 ± 1.9	−90.4 ± 3.7	5.0 ± 0.2	15.7 ± 2.5
p.Q2564R	0.19 ± 0.02	−44.9 ± 1.6	5.3 ± 0.2	1.2 ± 0.08 85.9 ± 2.6	11.9 ± 1.3 14.1 ± 2.6	−93.1 ± 1.8	5.2 ± 0.2	12.3 ± 1.1
Scrambled	0.18 ± 0.01	−40.4 ± 1.5	5.2 ± 0.3	1.1 ± 0.1 85.0 ± 3.0	9.1 ± 0.8 15.0 ± 3.0	−85.0 ± 2.4	5.0 ± 0.1	12.7 ± 2.9
si*Zfhx3*	0.18 ± 0.02	−40.0 ± 1.9	5.1 ± 0.3	1.1 ± 0.05 88.1 ± 0.7	9.3 ± 1.2 11.9 ± 0.7	−86.4 ± 1.9	5.1 ± 0.3	12.4 ± 3.1

A_finact_ and A_sinact_ = amplitudes of the fast and slow components of inactivation yielded by the fit of a biexponential function to the peak maximum current decay. τ_act_ = time constant of activation yielded by the fit of a monoexponential function to the peak maximum current. τ_finact_ and τ_sinact_ = fast and slow time constants of inactivation yielded by the fit of a biexponential function to the peak maximum current decay. τ_react_ = time constant of recovery from inactivation for I_Na_. V_hact_ and *k*_act_ = midpoint and slope values of conductance-voltage curves; V_hinact_ and *k*_inact_ = midpoint and slope values of the inactivation curves. Each value represents mean ± SEM of >6 cells/experiments from at least 3 different dishes in each group. Statistical comparisons were made by using ANOVA followed by Tukey’s test.

**Table 2 ijms-22-13031-t002:** Summary of all nonsynonymous exonic missense variants identified in the probands.

Proband	Gene	Genotype	Ancestral Allele /Variant	dbSNP_ID	MAF	Aminoacid Substitution	Transcript	Provean Prediction	SIFT Prediction	Polyphen Prediction	Mutation Taster	Mutation Assesor	LRT
*BrS-1*	*CACNA1B*	HET	G/A	rs774297154	0.0000041	R377Q	NM_000718.3	**Deleterious**	**Damaging**	Benign	Benign	-	-
	*FLNB*	HET	G/A	rs201369608	0.00007	R1009Q	NM_001164317.1	Neutral	Tolerated	Benign	**Probably deleterious**	**Low**	Neutral
	*ZFHX3*	HET	G/A	rs113497421	0.002	**V949I**	NM_006885.3	Neutral	Tolerated	**Probably damaging**	**Probably deleterious**	**Medium**	**Damaging**
*BrS-2*	*ANK2*	HET	A/G			D3126G	NM_001148.4	**Deleterious**	**Deleterious**	**Probably damaging**	**Deleterious**	**Medium**	**Damaging**
	*ZFHX3*	HET	A/G	rs141564201	0.0007	**Q2564R**	NM_006885.3	Neutral	Tolerated	Unknown	**Probably deleterious**	Neutral	
*AF-1*	*NEURL1*	HET	T/C			F35L	NM_004210.4	Neutral	Tolerated	Benign	**Probably deleterious**	**Low**	Neutral
	*ZFHX3*	HET	A/G	rs777360037	0.000004	**M1260T**	NM_006885.3	Neutral	Tolerated	Benign	**Probably deleterious**	**Low**	**Damaging**
*AF-2*	*ANK2*	HET	G/A	rs149963885	0.0006	E3016K	NM_001148.4	Neutral	Tolerated	Benign	Neutral	**Low**	Neutral
	*ZFHX3*	HET	A/G	rs777360037	0.000004	**M1260T**	NM_006885.3	Neutral	Tolerated	Benign	**Probably deleterious**	**Low**	**Damaging**

Only non-synonymous exonic missense variants with a coverage >30 and with an ocurrence in our local database = 1 were included. AF = atrial fibrillation; BrS = Brugada Syndrome; HET = heterozygous; LRT: likelihood ratio test; MAF = mean minor allele frequency from all ethnic groups where the variant was identified as provided in https://gnomad.broadinstitute.org/, accession on 29 November 2021.

## Data Availability

The raw data supporting the conclusions of this article will be made available by the authors upon reasonable request.

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
