# Peer review of "Zfhx3 Transcription Factor Represses the Expression of SCN5A Gene and Decreases Sodium Current Density (INa)"

_ijms, 2021, doi:10.3390/ijms222313031_

Round 1

Reviewer 1 Report

Major Comments

Dear authors,

This paper constitutes an interesting and well-supported document of the multi-level participation of the TF Zfhx3 in the control of cardiac excitability in human adult cardiac tissue.

Author Response

We would like to thank you for the time spent revising our MS and for your positive comments on it. All the changes included in the new version have been highlighted.

Reviewer 2 Report

In this paper, Rubio-Alarcón et al., nicely investigated effects of Zfhx3 on Nav1.5 current, protein expression and regulation. Using HL-1 they demonstrated that silencing Zfhx3 reduced peak INa and Zfhx3 significantly reduced transcriptional activity of human SCN5A, PITX2, TBX5, and NKX25 promoters. By having access to 3 families of patients with BrS or AF and Zfhx3 mutations, they tried to implicate those mutations in the arrhythmogenic phenotype. Nevertheless, the link seems independent of Nav1.5 or involved other variants. This paper is well written and propose a complex new regulatory pathway of Nav1.5.

Major

In figure 4 there is only 20-25% decrease of expression whereas there is more than 50% decrease in current amplitude. How can you explain that? Do the authors investigated localization of Nav1.5?

Statistical analysis: in figure 1 panel F scrambled values are all equal to 1. There is no SEM. Please to fix the mean at 1 and add a SEM for this condition as done in other figures. Otherwise the n in not enough to consider this condition as a population and not a sample. In the same panel, because there is 2 groups, ANOVA is not applicable as described in the legend. This point is already mentioned in the methods section but addition in figure legends is important to avoid mis-understanding. Please to use/mention the right test. In panel D there in only * whereas the legend said **. Same comments can be applied to all figures.

Figure 7 : BrS-1 proband has Zfhx3 and Cav2.2 mutations. hERG dans Cav2.2 mutants have not been searched in IV:1 and IV:2?

Figure 7E Please to add ANk2 variant as it has been done for panel A regarding KCNH2 and CACN1B genes

Figure 8A. Except for AF-1 variant, there is 2 variants for BrS-1 and Brs-2. Despite the fact that CACNA1B or ANK2 variants are not pathogenic, the authors can’t ignore that the phenotype of those patients might result from a combination of both mutations. Studying HL-1 with both mutants might be informative. This is an important point since by themselves overexpressing Zfhx3 mutants have no effects in Hl-1 cells (comparing to WT).

In figure 8A current density of M1260T and V949I mutants are very different to WT and Q2564R. This observation is not present in panel B. Do you have an explanation?

Regarding data with Zfhx3 mutants, do the expression of each variant is equal to WT? If not, that can change results interpretation.

Do Zfhx3 variants changed biophysical properties of Nav1.5? Ubiquitination?

Line 337 the authors stated that “Kv1.4, Kv1.5 and Kir3.4 channels” (and KATP) are affected by Zfhx3 silencing. Investigation of Kv currents with the mutations might be very informative to link mutants to arrhythmias.

Minor

The authors used GTEx database in order to demonstrate Zfhx3 expression in human samples. In order to increase the strength of the link between Zfhx3, NKX25, TBX5, PITX2 and SCN5A, correlations of expression might be informative.

For reproducibility which version of GTEx ?

Lines 47-49 => add reference. Same as 4 ?

Line 69 Scn5a

In figure 1 the authors could enlarge current traces. The last 20 ms of the step are not needed since they are not deadling with persistent current. In panel E it could be very informative to have also lanes for (-) and WT conditions in order to evaluate amplitude of overexpression of WT Zfhx3 in HL-1 cells. In the currents traces and I-V plot, current amplitude in Scrambled condition seems decreased compared to (-) condition. Is it significant?

Figure 1 panels C-D and E-F can be inverted.

Line 106 or not (-)

For activation curves is it I/Imax or G/Gmax ? Please to specify if it is G/Gmax

Figure 3 using the same scales between panels A-C and B-D can help comparison

Line 154 “Moreover, a concomitant decrease in the Zfhx3 mRNA expression was detected by Reverse 155 Transcription Quantitative PCR (RT-qPCR) experiments (P<0.05, n≥4) (Figure 4C).”. Panel C of figure 4 refers to SCN5A. Please to rephrase.

Line 208 NEDD4L has to be written in lower cases since it’s the endogenous gene from HL-1 cells.

Line 315 the authors said “expression of the cDNA encoding Zfhx3”. I would prefer to use “overexpression” since Zfhx3 is endogenously expressed in HL-1

Author Response

We would like to thank you for the time spent in revising our MS and for your insightful comments and suggestions that have greatly increased its quality. All the changes included in the new version have been highlighted. Please, find the attached file.

Round 2

Reviewer 2 Report

I would like to thank the authors for the changes which have increase the manuscript quality. Except the minor points listed below, I have no more requests.

Concerning data from patients and the link with newly identified mutations, the authors might add a limitation section in order to precise that their experimental conditions are not good enough to explain phenotype of patients. This is not an endpoint on this study related to regulation of Nav1.5 by Zfhx3. Further studies including co-variants might be already in progress.

In order to clarify difference between Nav1.5 expression and current, the authors argued that channel degradation should be promoted due to increase Nedd4-2 expression. Have the authors did western blots from biotinylation Nav1.5 (e.g. the membranous fraction) in order to validate this hypothesis? It is an important point to be included if the authors can do it easily.

The authors might add in the text, as they stated in the cover letter, that “Zfhx3 effects could impact the expression/function of scaffolding proteins such as SAP97 (PMID: 32612162) or α-syntrophin (PMID: 26786162) that interacts with Nav1.5 at different locations within the cardiomyocyte (e.g. intercalated disks and lateral membranes)”.

Figure 2 for reviewer 2 has to be included at least in supplemental data, this is an important result in order to avoid misinterpretation.

Author Response

I would like to thank the authors for the changes which have increase the manuscript quality. Except the minor points listed below, I have no more requests.

Thank you for the time spent in revising our MS again and for your comments.

Concerning data from patients and the link with newly identified mutations, the authors might add a limitation section in order to precise that their experimental conditions are not good enough to explain phenotype of patients. This is not an endpoint on this study related to regulation of Nav1.5 by Zfhx3. Further studies including co-variants might be already in progress.

Thanks for this comment. To precisely study the influence of the combined presence of the identified variants on the phenotype of each proband, it would have been necessary to generate cardiomyocytes derived from induced pluripotent stem cells (hiPSC-CM). In the new version we have included some comments about this in the Discussion.

Moreover, as described in Table 2, all three patients also carry other nonsynonymous exonic missense variants predicted as potentially deleterious. It is possible, that the phenotype of the patients is only apparent when, in addition to the variant in ZFHX3, the other variants of which they are carriers are present. Unfortunately, our experiments in HL-1 cells cannot rule out this hypothesis since it would have been necessary the generation of cardiomyocytes derived from induced pluripotent stem cells from each patient and the sequential correction of each variant with CRISPR-Cas9 technology.

In order to clarify difference between Nav1.5 expression and current, the authors argued that channel degradation should be promoted due to increase Nedd4-2 expression. Have the authors did western blots from biotinylation Nav1.5 (e.g. the membranous fraction) in order to validate this hypothesis? It is an important point to be included if the authors can do it easily.

We agree with you that the biotinylation assay would validate this hypothesis. Unfortunately, we did not conduct these experiments and we cannot perform them easily. In our experience, we would need at least two months to complete them. Following your suggestion, we have included some comments about it in the Discussion.

Furthermore, the INa inhibition exceeds the decrease in the total expression of Nav1.5 protein produced by Zfhx3. This would suggest that Zfhx3 ultimately decreases the presence of Nav1.5 channels in the cell membrane (not explored here) by additional translational or post-translational mechanisms that are currently unknown.

The authors might add in the text, as they stated in the cover letter, that “Zfhx3 effects could impact the expression/function of scaffolding proteins such as SAP97 (PMID: 32612162) or α-syntrophin (PMID: 26786162) that interacts with Nav1.5 at different locations within the cardiomyocyte (e.g. intercalated disks and lateral membranes)”.

Thank you for this comment. In the new version we have included some comments on this in the Discussion section (3.1 limitations of the study).

However, since HL-1 cells are not differentiated enough and do not display lateral membranes and intercalated disks as adult cardiomyocytes do, we were not able to analyze the possible effects of Zfhx3 on Nav1.5 localization. We surmise that Zfhx3 impacts the expression of the channels within the whole cardiomyocyte. However, we cannot rule out differential effects on Nav1.5 targeted to specific sub-domains since Zfhx3 may affect the expression/function of some of the multiple proteins that define distinct pools of Nav1.5 channels in cardiomyocytes [37,38].

Figure 2 for reviewer 2 has to be included at least in supplemental data, this is an important result in order to avoid misinterpretation.  

Following your recommendation, we included these data not only in the text as in the previous version (lines 284-286) but also in the Supplementary Figure 2
